# Capsaicinoids, Polyphenols and Antioxidant Activities of *Capsicum annuum*: Comparative Study of the Effect of Ripening Stage and Cooking Methods

**DOI:** 10.3390/antiox8090364

**Published:** 2019-09-02

**Authors:** Mansor Hamed, Diganta Kalita, Michael E. Bartolo, Sastry S. Jayanty

**Affiliations:** 1Department of Horticulture and Landscape Architecture, San Luis Valley Research Center, Colorado State University, Center, CO 81125, USA; 2Department of Horticulture and Landscape Architecture, Arkansas Valley Research Center, Colorado State University, Rocky Ford, CO 81067, USA

**Keywords:** pepper, capsaicin, dihydrocapsaicin, ascorbic acid total phenolics, antioxidant activity

## Abstract

Peppers (*Capsicum annuum* L.) are an important crop usually consumed as food or spices. Peppers contain a wide range of phytochemicals, such as capsaicinoids, phenolics, ascorbic acid, and carotenoids. Capsaicinoids impart the characteristic pungent taste. The study analyzed capsaicinoids and other bioactive compounds in different pepper cultivars at both the mature green and red stages. The effect of roasting on their nutritional content was also investigated. In the cultivars tested, the levels of capsaicin ranged from 0 to 3636 µg/g in the mature green stage and from 0 to 4820 µg/g in the red/yellow stage. The concentration of dihydrocapsaicin ranged from 0 to 2148 µg/g in the mature green stage and from 0 to 2162 µg/g in the red/yellow stage. The levels of capsaicinoid compounds in mature green and red /yellow stages were either reduced or increased after roasting depending on the cultivar. The ranges of total phenolic and total flavonoids compounds were 2096 to 7689, and 204 to 962 µg/g, respectively, in the green and red/yellow mature stage pods. Ascorbic acid levels in the peppers ranged from 223 to 1025 mg/ 100 g Dry Weight (DW). Both raw and roasted peppers possessed strong antioxidant activity as determined by 2,2-diphenyl-1-picrylhydrazyl) reagent (DPPH, 61–87%) and 2,2′-azino-bis (3-ethylbenzthiazoline-6-sulphonic acid) (ABTS, 73–159 µg/g) assays. Ascorbic acid and antioxidant activity decreased after roasting in the mature green and red stages, whereas total phenolics and flavonoids increased except in the mature green stage of Sweet Delilah and yellow stage of Canrio.

## 1. Introduction

Peppers are one of the most widely consumed food. They have diverse flavors, culinary uses, and nutritional content. After being introduced from the Americas, peppers have been incorporated into cultures and cuisines globally. Besides their direct culinary uses, peppers are also used for coloring, flavoring, preserving, nutraceutical, and medicinal purposes. Peppers belong to the genus *Capsicum. C. annuum, C. frutescens, C. chinense, C. pubescenes, and C. baccatum* are grown domestically or commercially [1,2]. Of these *C. annuum* is grown most extensively. 

Peppers are an excellent source of phytochemicals, such as anthocyanins, vitamins, phenolic acids, flavonoids, carotenoids, and capsaicinoids [3,4]. Various studies have demonstrated the benefits of bioactive compounds of peppers in vitro and in vivo. These compounds provide many nutritional and health benefits that include antioxidant, anti-inflammatory, and antimicrobial activities, reduced prevalence of type 2 diabetes and obesity, protection against hypercholesterolemia, and reduced prevalence of atherosclerotic cardiovascular diseases [5,6,7]. A recent study on the association of red hot chili consumption and mortality in a large American population observed a 13% reduction in mortality [8].

Capsaicinoids are the constituents in pepper that are responsible for pungency [9]. The degree of pungency is characterized in terms of Scoville heat units (SHU) measured based on the concentrations of capsaicinoid compounds within the fruit. SHU scale measures the number of times the extract is diluted to make pungency undetectable in sugar water [10]. Physiologically, capsaicinoids are synthesized by the condensation of vanillyl amine produced by the phenylpropanoid pathway and a branched-chain fatty acid produced by the catabolism of amino acids. Within the placental tissues of the developing pepper fruit, capsaicinoids are synthesized 20 to 30 days after pod formation and continue to accumulate as the fruit matures [11]. Some of the genes involved in the biosynthesis of capsaicinoids have been characterized in both pungent and nonpungent cultivars. Pun1 is a key regulator in the capsaicinoid pathway and controls the accumulation of capsaicinoids [12,13] Besides genetics, the concentration of capsaicinoids depends on other factors, such as stage of maturity and agronomic growing conditions [14,15,16]. Genetic diversity, agronomic practices and environmental conditions similarly influence the accumulation of polyphenolic compounds, minerals, Vitamin A, and ascorbic acid [17,18].

Numerous biochemical and physiological changes occur at different stages of pepper development due to changes in synthesis, transportation, and degradation of various metabolites [19,20]. At the mature green stage, the dominant pigments in peppers are chlorophylls and carotenoids. As maturation progresses, significant biochemical changes lead to the formation of new pigments (red/yellow carotenoids plus xanthophylls and anthocyanins). Moreover, the emission of volatile organic compounds that are associated with increased respiration, protein synthesis, formation of pectins, conversion of chlorophylls, and changes that include taste and flavor happens at the later stages of maturation and ripening [21]. Characterization of phytochemical changes in peppers that occur during maturation is essential since they could affect antioxidant activities, aroma, taste, postharvest storage, and ultimately consumer preference. 

Unique growing and environmental conditions including elevated solar radiation and significant shifts in diurnal temperatures have purported benefits for the development of unique flavor attributes of fruits and vegetables [21,22]. The total acreage of peppers grown in southern Colorado is about 950 acres with a farm gate value of roughly 5.7 million dollars (personal communication Dr. Bartolo). The biochemical composition of Colorado-grown peppers, which could impart unique aroma, nutritive, and medicinal properties has never been studied. As a result, one of the primary aims of our studies was to evaluate the capsaicinoids, total phenolics, and ascorbic acid and antioxidant activities of different *C. annuum* cultivars.

Peppers are consumed and processed in many forms. They are consumed raw in salads as well as blended into juices with other fruits and vegetables depending on consumer preferences. In addition to the fresh consumption, processors can dehydrate, pickle, cook, or roast peppers prior to consumption. Few reports have examined how heating peppers via cooking or roasting affects their phytonutrient content [23,24,25,26].

In Colorado and many other states, peppers are traditionally consumed after being roasted. Our is to investigate the changes in phytonutrients after roasting and the potential interaction with the stage of development and cultivar.

## 2. Materials and Methods 

### 2.1. Chemicals

Capsaicin, dihydrocapsaicin, ascorbic acid, Folin Ciocalteu reagent, sodium carbonate, gallic acid, potassium chloride, sodium acetate, 2,2-diphenyl-1-picrylhydrazyl) reagent (DPPH), 2,2′-azino-bis (3-ethylbenzthiazoline-6-sulphonic acid) (ABTS), potassium persulfate, trolox, quercetin, and all other reagents were purchased from Sigma-Aldrich (St. Louis, MO, USA).

### 2.2. Pepper Cultivars

All pepper samples in this study were collected from Colorado State University Arkansas Valley Research Center in Rocky Ford (AVRC) during the 2016 season at different pepper pod stages of green and red. Eighteen cultivars of *C. annum and C. chinense* species were field grown under commercial production conditions (Figure 1). Average high temperatures of Rocky Ford area in the months of July, August and September 73.8, 78.1 86.3 °F and minimums were 61.4, 59.0 and 52.2 °F respectively. Photosynthetically active radiation for those three months were 609, 482.6 and 425 W/m^2^, respectively. The elevation of Rocky Ford is 4180 feet above sea level. Three to five pepper pods of each cultivar were harvested from separate plants and were washed under running tap water and dried with paper towels. The peppers pods were cut into small pieces without peduncles, freeze-dried (LABCONCO, Kansas City, KS, USA), and ground to a fine powder by using a kitchen coffee grinder (Cuisinart). All sample were stored at −20 °C until further analysis.

### 2.3. Roasting 

Five to six peppers pods were washed and dried and cut into small pieces without peduncles, placed in an oven tray, transferred to the preheated oven set at 150 °C, and roasted for 20 min in a commercial oven. The oven was preheated for uniform heat distribution. All sample pieces were cooled, freeze-dried, and then ground and stored as described above. 

### 2.4. Extraction and Analysis of Capsaicinoid Compounds 

Five hundred milligrams of a ground sample from 3 to 5 pepper pods was added to a 15 mL polypropylene tube. Extraction and quantification of capsaicinoid compounds were performed essentially as described by Collins et al. [27]. Ten milliliters of methanol was added to each sample and kept in an orbital shaker overnight at 25 °C. The supernatant was transferred to a fresh 15 mL tube. Ten milliliters of methanol was added to the residue and shaken as just described. Then two supernatants were combined. One milliliter of the methanolic extract was filtered through 0.45 µm filter cartridge (Advanced Microdevices, Ambala, India) and put in a 1.8 mL sample glass vial for high-performance liquid chromatography (HPLC) analysis.

Capsaicin and dihydrocapsaicin were quantified using a Waters HPLC system equipped with a fluorescence detector and a Waters, Nova-Pak C18 4µm, 4.6 × 150 mm C18 column. Aqueous methanol A (10% methanol) and B (100% methanol) were used as eluent with a flow rate of 0.4 mL/min, and a gradient of 0 to 10 min, 80% A and 20% B. The fluorescence detector was set to an excitation wavelength of 280 nm and an emission wavelength of 338 nm. Levels of capsaicin and dihydrocapsaicin were estimated using a calibration curve with a standard of capsaicin and dihydrocapsaicin with concentrations ranging from 0.1 to 10 μg/mL. The resulting linear coefficient constants were 0.995 and 0.997, respectively.

### 2.5. SHU Determination 

The concentration (ppm) of capsaicin and dihydrocapsaicin compounds were converted into SHU (ppm) using their coefficient of the heat value with the following formula [10]: SHU = (capsaicin × 16.1) + (dihydrocapsaicin × 16.1)

### 2.6. Extraction of Phenolic Compounds

Phenoliccompounds were extracted from the freeze-dried material using methanol as the solvent. Five hundred milligrams of pepper powder from 3 to 5 pepper pods was mixed with 20 mL methanol in a 50 mL centrifuge tube and homogenized for 2 min followed by 2 min of vortexing. The mixture was incubated overnight at 25 °C. The supernatant was transferred to another test tube, and re-extractions were done with the residue. The two supernatant liquids were combined and filtered through a 0.45 μm filter and stored at −20 °C until further analysis.

### 2.7. Analysis of Total Phenolics (TP) and Total Flavonoids (TF) 

The TP content in the extract was determined using the Folin-Ciocalteu method, as described by Kalita et al. [28]. To measure the TP, 30 μL of the extract was mixed with 50 μL distilled water in wells of a 96-well plate. Fifty microliters of Folin Ciocalteu reagent and 80 μL Sodium carbonate (75 g/L) was added to each well in the plate, mixed well with a pipette, and then shaken for 4 min in a plate reader. The plates were incubated for 2 h at 25 °C in the dark. The absorbance of the contents was measured with a Power wave XS2 plate reader (BioTek) at 760 nm. Gallic acid was used as the standard and TP were quantified as μg/g of gallic acid equivalent per gram freeze-dried sample.

To measure TF (30 μL) was added to 80 μL aluminum chloride (20 g/L) in a 96-well at-bottom microplate on ice. Samples were shaken for 30 sec. and then the plates were kept in the dark at 25 °C for 1 h. The absorbance of the reaction was measured at 415 nm. Quercetin was used as the standard. TF were expressed as μg of quercetin equivalents per gram of freeze-dried weight. 

### 2.8. Extraction and Analysis of Ascorbic Acid 

Ascorbic acid was extracted from ground samples using meta-phosphoric acid and estimated using a method described by Watada et al. [29]. Five hundred milligrams of freeze-dried powder was mixed with 10 mL of 2.5% meta-phosphoric acid in a 15 mL tube. Samples were centrifuged at 5 rpm for 15 min. The supernatant was collected and filtered through a 0.45 µm filter paper. The quantification of ascorbic acid was performed using a Waters 2695 HPLC system (Waters Corporation, Milford, MA, USA) equipped with a Photodiode Array Detector (PDA) and a C18 column. The flow rate was 0.4 mL/ min with gradients of A (2.5% meta-phosphoric acid, 98% methanol) and B (100% methanol). The PDA was set at an excitation wavelength of 254 nm. Ascorbic acid standards were prepared in the range of 0.1 to 10 μg/mL in meta-phosphoric acid. The concentration of ascorbic acid in unknown samples was calculated from the standard curve. 

### 2.9. Antioxidant Activity

#### 2.9.1. DPPH Assay 

Radical scavenging activity of the extracts was evaluated using the scavenging activity of the stable DPPH free radical, which was measured as described by Kalita et al. [28] with slight modification. In a 96-well microplate, 30 μL of sample extract was added to each well containing 20 μL of distilled water. Two hundred microliters of 60 mgL^−1^ DPPH radical solution was added and mixed thoroughly. Samples were kept in the dark for 30 min. The absorbance of this reaction was measured using an ultraviolet spectrometer at 515 nm. A control was prepared by adding 30 μL of methanol without sample extract. The DPPH radical-scavenging activity was calculated using the following formula: DPPH radical-scavenging activity (%) = [(Acontrol − Asample/ Acontrol)] × 100
where A is the absorbance at 515 nm.

#### 2.9.2. ABTS Assay 

The ABTS radical cation-scavenging activity of the extracts was measured according to the method described by Kalita et al. [28] with modifications. ABTS radical (8 mM) was prepared and mixed with (3 mM) potassium acetate and the solution was kept in the dark for 12 h. The absorbance of the solution was adjusted to 1 then 285 µL of ABTS solution was added to 15 µL sample extract in the wells of a 96-well plate. The absorbance of the solution was recorded at 734 nm. Trolox was used as a standard to determine the antioxidant capacity in pepper samples, and the antioxidant capacity was expressed as µmol TE/g freeze-dried sample.

### 2.10. Statistical Analysis

Capsaicin, dihydrocapsaicin, total phenolics, total flavonoids, and antioxidant activities of pepper samples were conducted in triplicate and the results were reported as mean ± standard deviation (SD) values. The significant differences among the means were determined with one-way analysis of variance (ANOVA) by using SPSS version 12 (SPSS Institute, Chicago, IL, USA) at α = 0.05. Pearson’s correlation test was used to assess correlations between the means.

## 3. Results and Discussion

### 3.1. Levels of Capsaicinoid Compounds 

Capsaicinoid compounds are responsible for the pungency and unique taste in pepper cultivars. Twenty-three capsaicinoid analogs have been described [9]. These include capsaicin, dihydrocapsaicin, nordihydrocapsaicin, nordihydrocapsaicin, homodihydrocapsaicin, homocapsaicin, norcapsaicin, and nornorcapsaicin. Capsaicin and dihydrocapsaicin constitute 90% of these compounds in pepper. They are responsible for pungency and induce the sensation of hotness. Nor-dihydrocapsaicin has little effect on sensory attributes. We could detect the presence of capsaicin and dihydrocapsaicin using HPLC in select *C. annuum* and *C. chinense* cultivars. The separation and identification of these compounds are shown in Figure 2. 

We quantified the levels of capsaicin (Figure 3a) and dihydrocapsaicin (Figure 3b) in each cultivar.

In 16 pepper cultivars of *C. annuum*, the content of capsaicin ranged from 26 µg/g (Serano Mild) to 867 µg/g (CSU 243) where as dihydrocapsaicin ranged from 13 µg/g (Serano Mild) to 489 µg /g (Mosco) in the green stage. Similarly, in the red stage capsaicin content ranged from 49 µg/g (Serano Mild) to 819 (CSU 243) where as dihydrocapsaicin ranged from 14 µg/g (Serrano Mild) to 387 µg/g (CSU RLC) (Table 1) in 16 pepper cultivars of *C. annuum*. We could not detect capsaicin and dihydrocapsaicin in four cultivars Flavorburst, Canrio, Sweet Delilah and Aristotle.

Habanero is a popular cultivar grown in the Rocky Ford area of southern Colorado and is known for its high Scoville heat units. *Capsicum Chinense* species including Habanero was known to be the highly pungent chili pepper [30]. As expected, the highest levels of capsaicin and dihydrocapsaicin were detected in the Habanero variety (4820 and 2162.22 µg/g, respectively). The lowest levels of capsaicin and dihydrocapsaicin were found in the Serrano variety (26 and 13 µg/g respectively). There were significant differences among the cultivars of peppers (*p* value is set at ≤ 0.05). The findings from several studies have suggested that variations in capsaicinoid quantity can be attributed to intrinsic genetic factors of each cultivar or to the environmental conditions where they are cultivated [9]. Islam et al. [14] reported variation in capsaicinoid levels ranging from 0.02 to 72.05 mg/g in 139 different landraces of Capsicum. In addition to the cultivar type, accumulation of capsaicin and dihydrocapsaicin is affected by the activity of capsaicin synthase and peroxidase enzymes. The differences in the content of capsaicin are due to gene modifying factors that contribute to the accumulation of capsaicin and dihydrocapsaicin in different cultivars [31]. *C. chinense* displayed the highest level of capsaicin compared to *C. frutscens* and *C. annuum*, which was correlated with the greater expression of the capsaicin regulator gene, *Pun1* [9]. 

SHU is the crucial measurement to evaluate the pungency of pepper cultivars. Weiss [32] classified the pungency of peppers into five SHU levels depending on SHU: Non-pungent (0–700 SHU), mildly pungent (700–3000 SHU), moderately pungent (3000–25,000 SHU), highly pungent (25,000–70,000), and very highly pungent (>80,000 SHU). Based on this scale in our study most of the *C. annuum* cultivars, such as CSU 243, Fresno, CSU 256, Anaheim 118, Mosco, CSU RLC, CSU 290, Numex Joe Parker, Pueblo Chile, CSU 274, CSU 321, and Serrano mild peppers, could be grouped as moderately hot peppers. As expected, Habanero was classified as very hot. Sweet Dahlia, Flavorburst, Canario, and Aristotle were non-pungent peppers (Figure 4a,b).

Levels of capsaicinoid compounds change with maturation stage [4]. The capsaicinoid and dihydrocapsacionoids levels increased at the red stage (Figure 3a,b). Bae et al. [33] reported that Cayenne peppers displayed significant changes in the capsaicin and dihydrocapsaicin levels from 14.95% to 21.17%, and from 7.20% to 11.46%, respectively, at the green and red stages. A similar observation was obtained for Shimmatogarashi peppers, where the levels of SHU increased from 46,736 to 57,995 as fruit matured to the orange and red stages [19]. 

Sarpras et al. [9] grouped 136 capsicum germplasm belonging to *C. Chinense*, *C. frutescens*, and *C. annuum* into species having 0–10000, 10000–0.1 million, 0.1–0.3 million, 0.3–0.6 million, 0.6–0.9 million, and 0.9–1.2 million SHU.

The interactions between cultivars, stages and SHUs were summarized in Figure 5. The interaction plot, shows that the lines of the two stages are not parallel, that means there is a statistically significant interaction between cultivars and stages except in Serrano Mild. CSU 321, CSU 290 and CSU 243 cultivars. The differences between some cultivars are not evident because they are masked by the higher values of Habanero cultivar.

### 3.2. Levels of Bioactive Compounds (Ascorbic Acid, TP, TF, and TA)

Ascorbic acid plays a vital role as an antioxidant compound. It is very abundant in fruits and vegetables, particularly in peppers. Ascorbic acid levels in peppers varying depending on the cultivar and agro-climatic conditions. The content of ascorbic acid of the pepper cultivars we studied ranged from 222.55 (CSU 256) to 945.36 (Fresno) mg/100 g DW in the green stage from 314.87 (Mosco) to 752.54 (CSU 256) mg/100 g DW in the red stage (Table 1). A wide range of ascorbic acid has been reported in a number of pepper cultivars, indicating that the differences are related to cultivar, genetics, ripening stages, and agro-climatic conditions [4,34,35]. Mozafar et al. [36] suggested that the higher level of ascorbic acid in the matured stage was due to the light intensity and glucose level, which are the precursors of ascorbic acid. 

Peppers are an excellent source of bioactive compounds including anthocyanins, and flavonoids [37]. Phenolic compounds are secondary plant metabolites that play an essential role in antioxidant activity. TP in the selected peppers ranged from 2096 to 5578 µg/g in the green stage with the lowest levels in Serrano and the highest levels in Habanero. In the red stage, the levels of TP ranged from 3670.50 to 7689 µg/g with the lowest level in Pueblo chili and the highest levels in Serrano Mild. Interestingly, Serrano accumulated the most TP in the red stage. In general, the red matured stage displayed a higher level of TP than the green stage [19]. Earlier reports suggested that ripening of fruits and vegetable is associated with the significant accumulation of TP [38]. 

Anthocyanins are a subgroup of orange, purple, and red colored flavonoid compounds that are present in many fruits and vegetables [39]. Habanero‘s and Flavorburst’ orange-yellow colored peppers are rich in pelargonidin [40]. Flavonoids have antioxidant activity. Presently, the TF content of the pepper cultivars ranged from 204.44 (Habanero) to 756 (Flavorburst) µg/g in the green stage and from 557.28 (CSU 321) to 962.71 (Numex Joe E Parker) µg/g in the red stage (Table 1). TF was the lowest in the green stage of Habanero, was intermediate in Fresno (green stage), and the highest in red stage Numex Joe E. Parker. TF was the lowest in the red stage Mosco. A similar variation of flavonoids among different cultivars has been previously described [4]. 

### 3.3. Antioxidant Activities 

The use of DPPH is a standard means of measuring the antioxidant capacity of fruits and vegetable extracts. The DPPH scavenging activities of pepper cultivars are shown in Table 1. Scavenging of DPPH free radicals ranged from 61% to 87% and from 59% to 87% in green and red/yellow stages of different pepper cultivars, respectively. The highest antioxidant potential in green and red stages was observed in Canrio. Antioxidant potential was the lowest in Aristotle at the green stage and Mosco at the red stage. The difference in the antioxidant activities reflected the nature and level of antioxidant compounds found in the peppers. The use of ABTS is another way to measure antioxidant capacity. The ABTS radical cation scavenging activity of different kinds of pepper cultivars ranged from 72 to 157 µmol Trolox/g (Table 1). Among the *C. annuum* cultivars, CSU 290 had the highest 3-ethylbenzthiazoline-6-sulphonic acid (ABTS) scavenging capacity, while Fresno had the lowest in green stages. In the matured stage, Habanero had the highest ABTS scavenging capacity and Fresno had the lowest antioxidant potential. Prior descriptions of the antioxidant activities of peppers have indicated substantial antioxidant activities in the green to red stages of all peppers using DPPH, ABTS, or oxygen radical absorbance capacity (ORAC) assays [19,25]. However, most of the studies demonstrated that radical-scavenging activity increases as the fruit matures. Sora et al. [41] reported that the ABTS scavenging activities by pepper seed and pulps ranged from 89.25 to 141.25 µmol TE/g for seed extracts and from 17.17 to 97.40 µmol TE/g for pulp extracts. 

### 3.4. Correlation of TP, TF, TA, and Antioxidant Activities

Ascorbic acid, TP, and TF are the major compounds associated with antioxidant activity. Pearson correlation analysis of antioxidant activities and these bioactive compounds was carried out (Table 2). Capsaicin and dihydrocapsaicin had a positive correlation in all the pepper cultivars. However, no correlations were seen for capsaicinoids and the TP, TF, and ascorbic acid bioactive compounds. The correlation data suggested a relatively weak positive correlation of TP and TF with DPPH and ABTS antioxidant activities. TP displayed the highest correlation (*r* = 0.55) with the DPPH antioxidant assay. TF also show a positive correlation with a very poor correlation factor. These results contrasted with the previous report of very strong positive correlations of capsaicinoids, TP, and TF compounds with antioxidant activities [33]. Pearson correlation between the two methods of determining antioxidant activities were also low, but positive (*r* = 0.17). ascorbic acid displayed a very poor positive correlation with the DPPH assay, while it was negatively correlated with the ABTS assay. Manikharda et al. [19] reported a strong correlation between DPPH scavenging activities and TP, capsaicin, and ascorbic acid, with *r* = 0.9 in Shimatogarashi (*C. frutescenes*). Another study had similar findings [7]. However, in some cases, weak correlations were also found with the ORAC assay.

### 3.5. Effect of Roasting on the Levels of Capsaicin, Dihydrocapsaicin, TP, TF, and Antioxidant Activity

Cooking has a critical role in the compositional changes of peppers [25]. Due to the morphological and physiological differences among the different pepper genotypes, the changes in nutrients vary with cultivars [42]. Significant changes were evident in the levels of capsaicinoid compounds after roasting the peppers (Table 1). In most cases, the levels of capsaicin and dihydrocapsaicin were reduced after roasting the peppers at the green stage, except for CSU RLC, Fresno, and Numex Joe E. Parker (Table 1). A loss of capsaicinoids was evident in CSU-243, CSU-RLC, and Serrano Mild after roasting the peppers at the red stage, while the levels of capsaicinoids were reduced in CSU-243, CSU RLC, CSU-290, Pueblo chili, Mosco, and Fresno. These discrepancies in the levels of capsaicinoid compounds after cooking of cultivars might be due to their difference in the thickness of skin and physiological changes during ripening which could affect the heat permeability to the fruit materials. Previous studies on the effect of cooking on capsaicinoids also indicated contradictory results. Srinavasan et al. [43] and Topuz et al. [44] reported that there is a loss of 0% to 30% and 215% to 100% of the capsaicinoids in Indian Thai, and Turkish peppers after cooking. Orneals-Paz et al. [26] reported that moderate loss of capsaicinoids were observed after boiling in Mexican peppers, while grilling enhanced the level of capsaicinoid compounds. Heat treatment during cooking disrupts the pepper cell wall and could affect the extractability of these compounds.

There was a significant reduction in ascorbic acid after roasting in both green and red stage peppers, ranging from 8% to 80% irrespective of the stage. Similarly, loss of ascorbic acid was observed by Howard et al. [45] in pungent peppers (Jalapeno) on heat treatment. Ascorbic acid disappears significantly after cooking due to their thermolability and solubility in water. Lawermmar et al. [46] studied the effect of cooking on ascorbic acid content in pepper, green peas, spinach, pumpkin, and carrots. The highest loss of ascorbic acid (64.71%) was seen in peppers after 30 min. Loss of ascorbic acid in cooked peppers is due to the thermal oxidation of ascorbic acid to dehydroascorbic acid followed by hydrolysis to 2, 3 diketogluconic acid and conversion to other polymeric compounds [47]. Yadav and Shegal [48] reported that cooking at high temperature for a long time leads to pronounced atmospheric oxidation of food constituents. Chuah et al. [23] described inconsistencies in ascorbic acid content after boiling of green and red peppers, suggested that the differences were due to the thickness of the pepper fruit skin. The thinner cell membrane would be more permeable to heat, which could result in the rapid leaching of ascorbic acid from the peppers. The effect of roasting on TA compounds is shown in Table 1.

Interestingly, in all roasted peppers the phenolic compounds increased by 1% to 106% in both green and red stages compared to fresh peppers. Since there is no biosynthesis of TP occurs after harvesting/roasting of peppers, a higher level of TP might be due to better extractability from the roasted peppers [24]. Several studies reported that cell disruption in peppers increases the leaching of compounds into the solvents and increases the level of phenolic compounds compared to uncooked peppers. Shaimaa et al. [49] reported that phenolic and flavonoids compounds were increased by cooking treatment of some sweet and chili pepper. However, José de Jesús et al. [24] suggested that the effect of cooking on peppers compounds could cause either an increase or decrease in TP. Turkmen et al. [50] reported that boiling, steaming, and microwaving increased the phenolic content. Similar results were suggested by Orneals-Paz et al. [26] boiling and grilling cooking enhanced the levels of TP. Contradictory reports suggested that cooking methods do not influence the phenolic content due to the inactivation of polyphenol oxidase enzyme by heat. Chuah et al. [23] reported a significant loss of phenolic compounds from colored bell peppers during cooking.

There was a reduction of antioxidant activity in cooked peppers in both the DPPH and ABTS assays, even though increased levels of antioxidants were evident after cooking due to better extractability. The antioxidant potential is a synergistic property of all the antioxidant compounds, and it depends on the nature of the compounds. After cooking, there could be a change or modification in the chemical properties of the antioxidants affecting radical-scavenging activities. 

## 4. Conclusions 

Colorado pepper cultivars are good sources of phytonutrients viz capsaicinoids, ascorbic acid, phenolic compounds, such as flavonoids. The high level of these nutrients is retained in peppers even after roasting, as is pronounced antioxidant activity. The present information will be helpful for breeders to select better parents to develop new pepper cultivars with the desired taste and pungency with health-promoting compounds. 

## Figures and Tables

**Figure 1 antioxidants-08-00364-f001:**
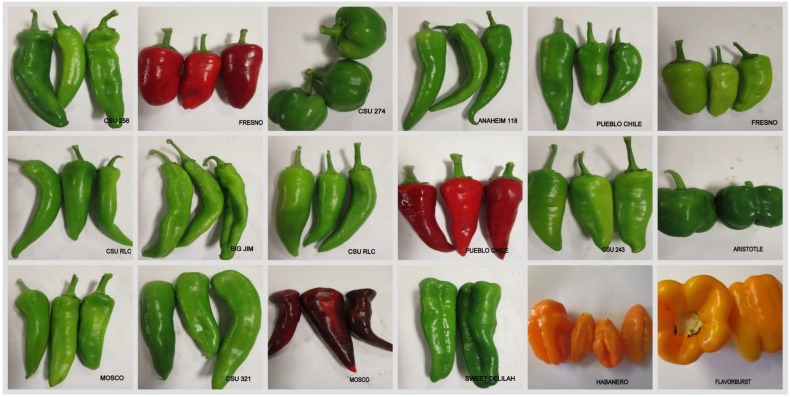
Morphological diversity of pepper pods of selected cultivars developed and grown at Arkansas Valley Research Center, Rocky Ford. All cultivars belong to *C. annuum* species except Habanero which is a *C. chinense*.

**Figure 2 antioxidants-08-00364-f002:**
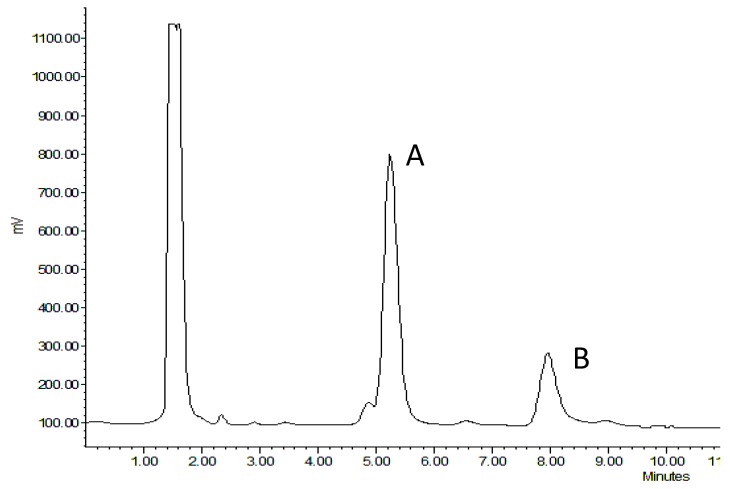
A representative high-performance liquid chromatography (HPLC) chromatogram of *Capsicum annuum* cultivar Colorado State University (CSU) 256 green showing baseline separation of (**A**) capsaicin, and (**B**) dihydrocapsaicin using Waters, Nova-Pak C18 column using fluorescence detector. Peaks were identified by comparing retention times to those of standard compounds (capsaicin and dihydrocapsaicin).

**Figure 3 antioxidants-08-00364-f003:**
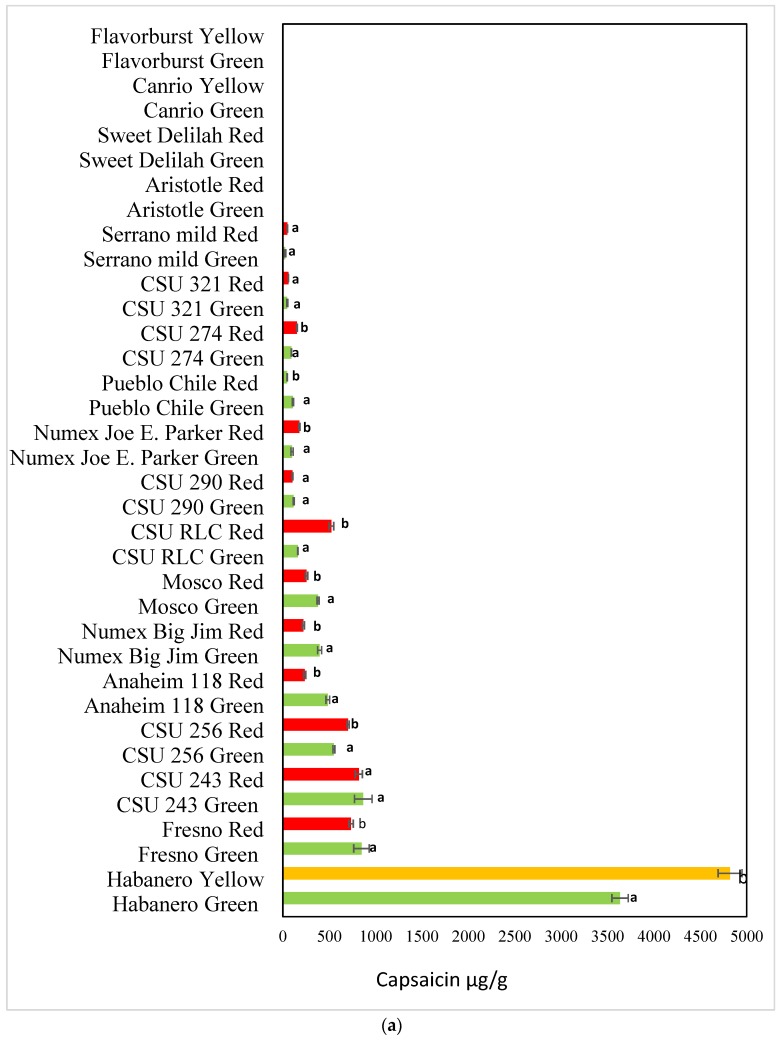
(**a**) Comparison of capsaicin levels in *C. annuum cultivars and Habanero*. Data are the mean of three replicates with standard deviation and are expressed as per gram freeze-dried weight. Significant differences are denoted by different letters, while the same or shared letters indicate that they are not significant to each other. (**b**) Comparison of dihydrocapsaicin levels in *C. annuum cultivars and Habanero*. Data are the mean of three replicates with standard deviation and are expressed as per gram freeze-dried weight. Significant differences are denoted by different letters, while the same or shared letters indicate that they are not significant to each other.

**Figure 4 antioxidants-08-00364-f004:**
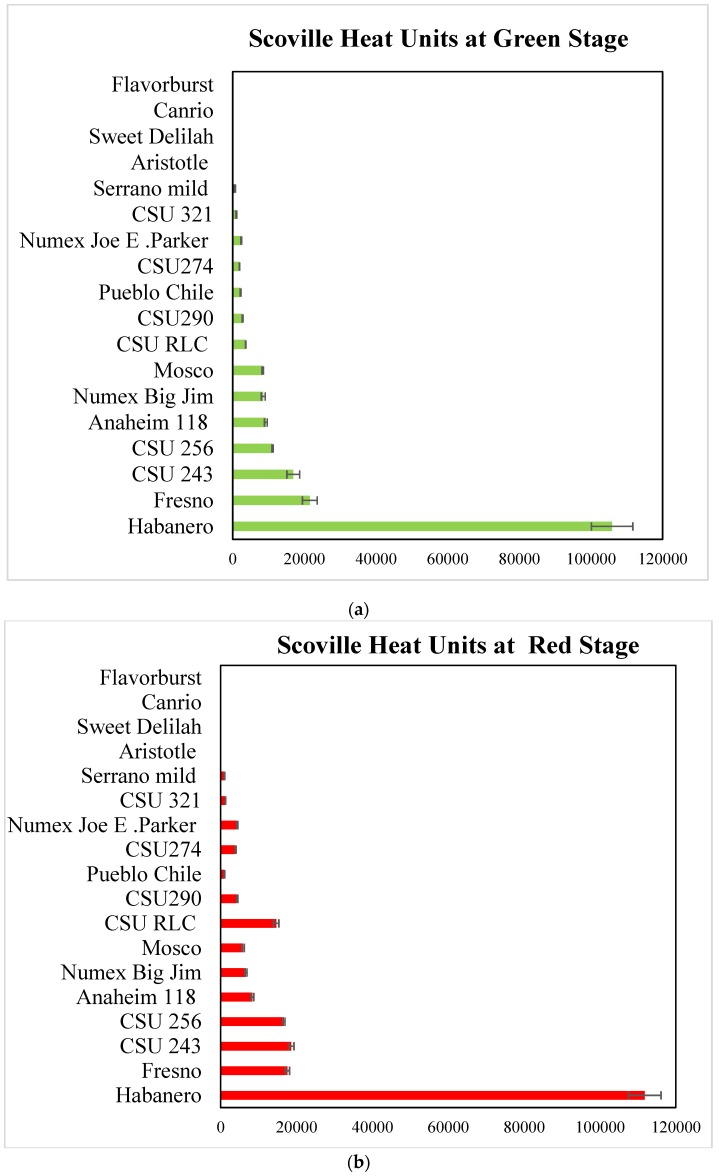
(**a**) Total capsaicinoids content estimated in *C. annuum* and *C. Chinense* genotypes in green pepper pods in Scoville heat units (SHU). (**b**) Total capsaicinoids content estimated in *C. annuum* and *C. Chinense* genotypes in red pepper pods in Scoville heat units (SHU). The significance level was set at *p* < 0.05. Bars represent stand deviation (SD).

**Figure 5 antioxidants-08-00364-f005:**
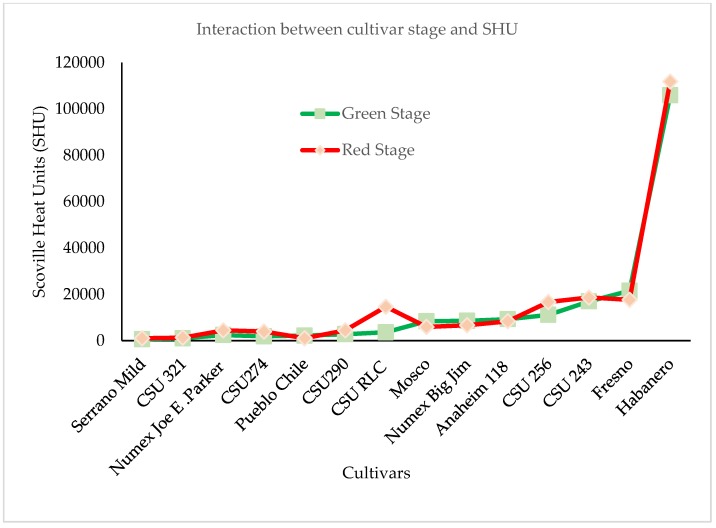
The interaction between cultivar and stage for SHU using Excel ver 2016. Cultivars that did not show capsaicin and dihydrocapsaicin were not included in this analysis.

**Table 1 antioxidants-08-00364-t001:** Change in content of capsaicinoid compounds in pepper pods as a function of ripening stage and cooking. Effect of roasting on the nutrient contents in *C. annuum* and *C. Chinense* viz. habanero.

Cultivars	Maturation Stage	Capsaicin (µg/g)		Dihydrocapsaicin (µg/g)		Vit.C (mg/100 g DW)		Total Phenolic (µg/g)		Total Flavonoids (µg/g)		Antioxidant Activity			
												DPPH (%)		ABTS (µg/g)	
		Raw	Roasted	Raw	Roasted	Raw	Roasted	Raw	Roasted	Raw	Roasted	Raw	Roasted	Raw	Roasted
Flavorburst	Green	UDL	UDL	UDL	UDL	401 ^a^	163^↓^	4279 ^a^	6749^↑^	756 ^a^	554^↓^	76 ^a^	55^↓^	129 ^a^	86^↓^
	Yellow	UDL	UDL	UDL	UDL	478 ^b^	275^↓^	5398 ^b^	6183^↑^	794 ^b^	671^↓^	87 ^b^	70^↓^	139 ^a^	123^↓^
Canrio	Green	UDL	UDL	UDL	UDL	693 ^a^	328^↓^	5578 ^a^	5783^↑^	500 ^a^	697^↑^	87 ^a^	61^↓^	110 ^a^	85^↓^
	Yellow	UDL	UDL	UDL	UDL	1025 ^b^	315^↓^	6316 ^b^	6225^↓^	573 ^b^	798^↑^	86 ^a^	71^↓^	157 ^b^	103^↓^
Sweet Delilah	Green	UDL	UDL	UDL	UDL	420 ^a^	148^↓^	3314 ^a^	3226^↓^	547 ^a^	660^↑^	63 ^a^	50^↓^	113 ^a^	105^↓^
	Red	UDL	UDL	UDL	UDL	481 ^b^	250^↓^	5115 ^b^	6489^↑^	625 ^b^	880^↑^	72 ^b^	58^↓^	152 ^b^	82^↓^
Aristotle	Green	UDL	UDL	UDL	UDL	480 ^a^	180^↓^	2599 ^a^	3252^↑^	329 ^a^	405^↑^	64 ^a^	54^↓^	107 ^a^	93^↓^
	Red	UDL	UDL	UDL	UDL	418 ^b^	217^↓^	4729 ^b^	5318^↑^	556 ^b^	727^↑^	59 ^a^	41^↓^	156 ^b^	118^↓^
Serrano Mild	Green	26 ^a^	57^↑^	13 ^a^	12^↓^	243 ^a^	193^↓^	2096 ^a^	3899^↑^	415 ^a^	755^↑^	65 ^a^	52^↓^	72 ^a^	108^↓^
	Red	49 ^a^	71^↑^	14 ^a^	26^↑^	467 ^b^	82^↓^	7689 ^b^	8188^↑^	643 ^b^	887^↑^	75 ^b^	64^↓^	101 ^b^	98^↓^
CSU 321	Green	48 ^a^	19^↓^	16 ^a^	9^↓^	335 ^a^	175^↓^	2841 ^a^	3277^↑^	443 ^a^	636^↑^	66 ^a^	62^↓^	106 ^a^	76^↓^
	Red	60 ^a^	61^↑^	19 ^a^	21^↑^	648 ^b^	405^↓^	5074 ^b^	7117^↑^	557 ^b^	785^↑^	71 ^b^	55^↓^	126 ^b^	83^↓^
CSU 274	Green	92 ^a^	25^↓^	26 ^a^	10^↓^	327 ^a^	94^↓^	3165 ^a^	3537^↑^	484 ^a^	745^↑^	67 ^a^	31^↓^	146 ^a^	93^↓^
	Red	152 ^b^	188^↑^	94 ^b^	182^↑^	496 ^b^	245^↓^	5941 ^b^	7406^↑^	626 ^b^	865^↑^	78 ^b^	56^↓^	154 ^a^	128^↓^
Pueblo Chile	Green	108 ^a^	89^↓^	28 ^a^	27^↓^	337 ^a^	114^↓^	3208 ^a^	4232^↑^	695 ^a^	784^↑^	69 ^a^	58^↓^	75 ^a^	67^↓^
	Red	47 ^b^	45^↓^	16 ^a^	17^↑^	386 ^a^	231^↓^	3670 ^b^	5845^↑^	725 ^a^	844^↑^	79 ^b^	62^↓^	104 ^b^	85^↓^
Numex Joe E. Parker	Green	99 ^a^	225^↑^	50 ^a^	154^↑^	494 ^a^	105^↓^	3845 ^a^	5958^↑^	656 ^a^	984^↑^	70 ^a^	65^↓^	111 ^a^	77^↓^
	Red	177 ^b^	251^↑^	97 ^b^	130^↑^	333 ^b^	243^↓^	4360 ^b^	5019^↑^	962 ^b^	740	81 ^b^	68^↓^	144 ^b^	125^↓^
CSU 290	Green	117 ^a^	84^↓^	56 ^a^	29^↓^	643 ^a^	357^↓^	2769 ^a^	5730^↑^	586 ^a^	874^↑^	61 ^a^	69^↓^	155 ^a^	104^↓^
	Red	109 ^a^	87^↓^	19 ^b^	24^↑^	628 ^a^	358^↓^	4448 ^b^	6584^↑^	714 ^b^	745^↑^	82 ^b^	63^↓^	110 ^b^	97^↓^
CSU RLC	Green	162 ^a^	338^↑^	62 ^a^	130^↑^	252 ^a^	167^↓^	3893 ^a^	5783^↑^	619 ^a^	828^↑^	73 ^a^	56^↓^	152 ^a^	78^↓^
	Red	522 ^b^	446^↓^	387 ^b^	377^↓^	345 ^b^	144^↓^	5441 ^b^	7190^↑^	777 ^b^	865^↑^	75 ^a^	66^↓^	123 ^b^	95^↓^
Mosco	Green	379 ^a^	252^↓^	141 ^a^	84^↓^	567 ^a^	351^↓^	3417 ^a^	4350^↑^	415 ^a^	883^↑^	65 ^a^	51^↓^	94 ^a^	79^↓^
	Red	256 ^b^	254^↓^	112 ^b^	118^↑^	314 ^b^	232^↓^	3876 ^b^	4603^↑^	643 ^b^	804^↑^	75 ^b^	55^↓^	157 ^b^	74^↓^
Numex Big Jim	Green	398 ^a^	385^↓^	181 ^a^	138^↓^	410 ^a^	365^↓^	3341 ^a^	4392^↑^	423 ^a^	606^↑^	63 ^a^	65^↓^	152 ^a^	99^↓^
	Red	222 ^b^	297^↑^	212 ^b^	114^↓^	465 ^b^	240^↓^	6335 ^b^	7613^↑^	653 ^b^	753^↑^	82 ^b^	56^↓^	111 ^b^	106^↓^
Anaheim 118	Green	484 ^a^	454^↓^	90 ^a^	86^↓^	339 ^a^	211^↓^	3276 ^a^	6367^↑^	504 ^a^	508^↑^	80 ^a^	68^↓^	121 ^a^	102^↓^
	Red	235 ^b^	325^↑^	282 ^b^	148^↓^	391 ^b^	254^↓^	4676 ^b^	5580^↑^	641 ^b^	743^↑^	85 ^a^	69^↓^	150 ^b^	103^↓^
CSU 256	Green	550 ^a^	281^↓^	141 ^a^	75^↓^	223 ^a^	159^↓^	2758 ^a^	2795^↑^	498 ^a^	564^↑^	68 ^a^	40^↓^	85 ^a^	67^↓^
	Red	703 ^b^	912^↑^	332 ^b^	464^↑^	753 ^b^	205^↓^	5256 ^b^	6718^↑^	747 ^b^	741	78 ^b^	54^↓^	136 ^b^	112^↓^
CSU 243	Green	867 ^a^	512^↓^	183 ^a^	92^↓^	338 ^a^	79^↓^	4472 ^a^	5538^↑^	634 ^a^	708^↑^	61 ^a^	49^↓^	120 ^a^	95^↓^
	Red	819 ^a^	514^↓^	337 ^b^	249^↓^	370 ^a^	214^↓^	5186 ^b^	7950^↑^	704 ^b^	935^↑^	77 ^b^	60^↓^	145 ^b^	109^↓^
Fresno	Green	848 ^a^	729^↓^	489 ^a^	394^↓^	945 ^a^	464^↓^	3549 ^a^	4624^↑^	713 ^a^	807^↑^	85 ^a^	72^↓^	159 ^a^	121^↓^
	Red	735 ^b^	599^↓^	359 ^b^	331^↓^	366 ^b^	167^↓^	4527 ^b^	6444^↑^	792 ^b^	861^↑^	74 ^b^	67^↓^	137 ^b^	121^↓^
Habanero	Green	3636 ^a^	3834^↑^	2148 ^a^	1441^↓^	820 ^a^	160^↓^	4679 ^a^	5041^↑^	204 ^a^	838^↑^	74 ^a^	59^↓^	153 ^a^	98^↓^
	Yellow	4820 ^b^	4876^↑^	2162 ^a^	1572^↓^	349 ^b^	249^↓^	6505 ^b^	6703^↑^	467 ^b^	627^↑^	86 ^b^	68^↓^	155 ^a^	122^↓^

Values are expressed as actual values of compounds, loss (down arrow) or gain (up arrow) from the mean of three values. Levels of ascorbic acid, total phenolics (TP), total flavonoids (TF), and antioxidant activities. Data are mean of three replicates with standard deviation and are expressed as per gram freeze-dried sample. The comparison at a specific variety is between green and red (column direction) and raw and roasted (raw direction). Significant differences are denoted by different letters (^a,b^), while the same or shared letters indicate that they are not significant to each other. UDL: Under detection limit. DPPH: 2,2-diphenyl-1-picrylhydrazyl). ABTS: 2,2′-azino-bis (3-ethylbenzthiazoline-6-sulphonic acid). *p* value is set at ≤ 0.05.

**Table 2 antioxidants-08-00364-t002:** Pearson’s correlation coefficient analysis among capsaicin, dihydrocapsaicin, total phenolics, total flavonoids, and antioxidant activities.

Variables	Capsaicin	Dihydro Capsaicin	Total Phenolics	Total Flavonoids	Ascorbic Acid	AA1 (DPPH)	AA2 (ABTS)
Capsaicin	1						
Dihydro Capsaicin	1	1					
	<0.0001						
Total Phenolics	0.4464	0.4464	1				
	0.0173	0.0173					
Total Flavonoids	−0.3906	−0.3906	0.1474	1			
	0.0399	0.0399	0.4541				
Ascorbic Acid	0.1986	0.1986	0.2387	−0.0209	1		
	0.3109	0.3109	0.2212	0.916			
AA1 (DPPH)	0.2898	0.2898	0.5594	0.3389	0.0446	1	
	0.1347	0.1347	0.002	0.0777	0.8218		
AA2 (ABTS)	0.2981	0.2981	0.3292	−0.0695	−0.0758	0.1725	1
	0.1234	0.1234	0.0872	0.7253	0.7013	0.3801

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
