# Peer review of "Capsaicinoids, Polyphenols and Antioxidant Activities of Capsicum annuum: Comparative Study of the Effect of Ripening Stage and Cooking Methods"

_antioxidants, 2019, doi:10.3390/antiox8090364_

Round 1
Reviewer 1 Report
The paper has some information that could be useful to pepper breeders or to people searching for good sources of antioxidants. However, I found it a bit difficult to read because there were so many lists and few attempts to tie the data together in manageable pieces of information. I think that if the novelty of the paper lies in the focus on peppers grown in Colorado, and phenolic biosynthesis depends partly on environmental factors, it would be helpful to include weather data (photosynthetically active radiation, temperature, and precipitation) during the 2016 growing season so that readers can make comparisons to weather conditions in other areas where peppers are grown.
I am puzzled by the inclusion of the C. chinense cultivar. Given the variation in capsaicinoids and SHU among all the C. annuum cultivars, can conclusions be drawn about C. chinense capsaicinoids based on one cultivar? Perhaps a sentence is needed to explain why Habanero peppers are included in the study.
I am puzzled by the number of significant figures used in the table and the text. I was taught that the number of significant figures in a result cannot exceed the number of significant figures in the least accurate measurement used to get the result. Based on my experience with pipettes and syringes, it seems difficult to me to think that the least accurate measurement in the study had four or five significant figures.
The results and discussion seem to consist of lists of different cultivars and their antioxidant properties or phytochemicals. It would be helpful to the reader to see these lists grouped together in some way that makes it easier to follow the results. Are there main effects of maturity (red or yellow vs. green)? Of cooking treatment (raw vs. roasted)? Of cultivar? A table with this type of information would be relatively easy to read and retain. Are there interactions of maturity, cooking treatment, or cultivar? Graphs with that type of information would help to group things together and would help the reader to see which cultivars are similar, and which ones are different. Figure 3 does not specify which cultivars are different from each other, so some means separations and P-values would be helpful.
Line 37. C. annuum needs to be italicized.
Line 40. On lines 38-39, types of phenolic compounds are listed. When you say that “Polyphenolic compounds are abundant in peppers,” are you referring to another class of phenolic compounds, different from the ones listed on lines 38-39? Could you give an example? If polyphenolic compounds refers to the ones mentioned on lines 38-39, the phrase is redundant.
Line 56. Capsicum needs to be italicized.
Line 60. Ascorbic does not need to be capitalized.
Line 69. Change “are essential” to “is essential” (verb agrees with “Characterization”).
Line 72. Delete “in”.
Lines 74-76. Are peppers a major crop in Colorado? Information about the acreage of pepper cultivation in Colorado would be helpful.
Line 100. Change “sample” to “samples”.
Lines 116-117. I have never seen 0.45 nm filter paper. Could you please give the brand name and part number for this item? Do you mean 0.45 µm filters?
Line 130. How is the coefficient for the heat value obtained? A reference or brief explanation would be helpful to people unfamiliar with SHU.
Line 132. Flavonoids and anthocyanins are phenolic compounds, too. By “phenolics,” do you mean “phenolic acids”?
Line 147. Something appears to be missing at the beginning of this sentence. Do you want to start with “To measure the TF, extract (30 µL) was added…”?
Lines 158-159. These two lines almost repeat each other. Please check to see which part needs to be removed.
Line 160. I am uncertain that I understand “L X mol–1 & cm–1” Are the X and & signifying multiplication?
Line 179. Is mg-l written correctly? Should it be mg · L-1?
Line 203. Twenty three should be hyphenated (twenty-three).
Line 248. I would rewrite the beginning of this sentence as “In 16 pepper cultivars of C. annuum, the content of capsaicin ranged from …” (Take out “the that we studied”). What about the pepper cultivars with no capsaicin or dihydrocapsaicin? I think that it should be mentioned here that capsaicinoids were absent in some cultivars, instead of on lines 281-282, because that absence is very striking in the graphs.
Line 249. Whereas is one word. Also, the microgram abbreviation needs to be corrected (replace ug with µg).
Line 250. Take out “was”.
Line 280. Why was it expected that the Habanero variety would have the most capsaicin and dihydrocapsaicin? Is it because Habanero peppers are known to be hot? Some readers may not be familiar with Habanero peppers and need an explanation. Are there other reports of Habanero capsaicin and dihydrocapsaicin concentrations? If there are other reports, they should be cited.
Lines 283-284. When you say that there were significant differences among the cultivars, do you have a P-value that you can provide to support the statement? It is clear from the graphs that there was a broad range, but I think it is conventional to give a P-value in addition. Means separations and letter groupings would be helpful as well, to see which cultivars are most similar to each other.
Lines 284-286. The findings from the literature are nicely summarized, but a relationship to your findings is missing. If environmental conditions affect capsaicinoid concentrations, can you say something about the environmental conditions under which these were cultivated?
Lines 331-332. I thought that anthocyanins were polyphenols, or at least phenolic compounds. Could you include an example of a polyphenolic compound to clarify the difference between anthocyanins and polyphenolic compounds?
Line 340. Are anthocyanins ever orange? I would expect carotenoids to be responsible for an orange color. Can you give an example of an orange anthocyanin?
Lines 348-349. Change “of a fruits and vegetable extracts” to “of fruit and vegetable extracts”.
Line 441. Do you have a reference about the biosynthesis of TP occurring after roasting? How can dead cells continue to produce phenolics? I would expect the existing phenolics to become oxidized, or for the conjugated sugars to be hydrolyzed, but not for the cells to make more. A reference would be really helpful.
Lines 452-453. Did you do a statistical test to determine if the changes in antioxidant activity after roasting were significant? You might be able to find a group of peppers whose antioxidant activities did not change significantly.
Lines 453-454. How do you know that extractability after cooking was better? In lines 441-442, better extractability was suggested, but it did not seem to be something that had been tested.
Lines 458-460. High levels of ascorbic acid were not retained after roasting. The data show that ascorbic acid decreases to 25 to 50% of its original value after roasting. Antioxidant activity also decreased after roasting. Could you do statistical tests to determine if the changes were significant?
Line 459. Flavonoids are phenolic compounds. Perhaps you could say “and phenolic compounds such as flavonoids.”
Lines 462-464. The last 2 sentences seem to be part of a template.
Table 1. Some of these values contain 4 significant figures (e.g. 4876 µg/g capsaicin in roasted yellow Habanero). The maximum number of significant figures cannot exceed the number in your least certain measurement. It seems to me that it would be better to round a value like 4876 to 4880.
Table 1. Are the superscripted letters comparisons within a cultivar (red vs. green for each cultivar), or are they comparisons made across all the cultivars for a given measurement? That information should be specified in the table caption or a footnote.
Table 1. Are the numerical increases or decreases statistically significant? That would be a helpful piece of information.
Author Response
Dear Reviewer
Thanks for taking the time to review our manuscript.
Please find attached responses to your edits.
Thanks

Reviewer 2 Report
Minor modification is required.

Author Response
Dear Reviewer
Thanks for taking the time to review our Manuscript. I sincerely appreciate your effort.
Please see the detailed response to your corrections.
Thank You.
